# Rev Protein Diversity in HIV-1 Group M Clades

**DOI:** 10.3390/v16050759

**Published:** 2024-05-10

**Authors:** Aleksey Lebedev, Kristina Kim, Ekaterina Ozhmegova, Anastasiia Antonova, Elena Kazennova, Aleksandr Tumanov, Anna Kuznetsova

**Affiliations:** 1Gamaleya National Research Center for Epidemiology and Microbiology, 123098 Moscow, Russia; kimsya99@gmail.com (K.K.); belokopytova.01@mail.ru (E.O.); anastaseika95@mail.ru (A.A.); kazennova@rambler.ru (E.K.); desep@mail.ru (A.T.); 2Mechnikov Scientific Research Institute of Vaccines and Serums, 105064 Moscow, Russia

**Keywords:** HIV-1, rev, group m, subtypes, diversity, sequences

## Abstract

The HIV-1 Rev protein expressed in the early stage of virus replication is involved in the nuclear export of some forms of virus RNA. Naturally occurring polymorphisms in the Rev protein could influence its activity. The association between the genetic features of different virus variants and HIV infection pathogenesis has been discussed for many years. In this study, Rev diversity among HIV-1 group M clades was analyzed to note the signatures that could influence Rev activity and, subsequently, clinical characteristics. From the Los Alamos HIV Sequence Database, 4962 Rev sequences were downloaded and 26 clades in HIV-1 group M were analyzed for amino acid changes, conservation in consensus sequences, and the presence of clade-specific amino acid substitutions (CSSs) and the Wu–Kabat protein variability coefficient (WK). Subtypes G, CRF 02_AG, B, and A1 showed the largest amino acid changes and diversity. The mean conservation of the Rev protein was 80.8%. In consensus sequences, signatures that could influence Rev activity were detected. In 15 out of 26 consensus sequences, an insertion associated with the reduced export activity of the Rev protein, 95QSQGTET96, was identified. A total of 32 CSSs were found in 16 clades, wherein A6 had the 41Q substitution in the functionally significant region of Rev. The high values of WK coefficient in sites 51 and 82, located on the Rev interaction surface, indicate the susceptibility of these positions to evolutionary replacements. Thus, the noted signatures require further investigation.

## 1. Introduction

The HIV-1 Rev protein plays an important role in the viral life cycle, coordinating the nuclear export of unspliced and incompletely spliced virus mRNAs [1]. Three types of virus mRNAs are required for the production of HIV-1 virions: (1) fully spliced mRNAs, (2) full-sized, or unspliced, mRNAs, and (3) incompletely spliced mRNAs. In the earlier stage of virus replication, only fully spliced mRNAs, which encode the viral proteins Tat, Rev, and Nef, are transported in the cytoplasm for translation. Full-sized, or unspliced mRNAs, encode the structural polyproteins Gag and Gag–Pol and also serve as genomic RNA. The incompletely spliced RNAs encode one structural protein, Env, and three accessory proteins: Vif, Vpr, and Vpu. At first, underspliced (unspliced and incompletely spliced) mRNAs are recognized by the host cells as “immature” and are retained in the nucleus. However, underspliced mRNAs contain an RNA structure, known as the Rev response element (RRE). Rev, expressed from a fully spliced viral RNA, penetrates in the nucleus, binds to the RRE through oligomerization, and induces the export of underspliced mRNAs to the cytoplasm [1,2,3]. Rev also takes part in RNA splicing, stability, and translation; however, its influence on these processes remains poorly understood [1]. Rev is a ~13 kD protein composed of 116 amino acids. It is coded by two exons that overlap with other genes: *tat* and *env* [3,4] (Appendix A).

The domain organization of the Rev protein includes functional regions (OD: Oligomerization domain; ARM: Arginine-rich motif; NES: Nuclear export sequence) and linker regions (N-term: N-terminus; Turn; ONL: OD-NES linker; C-term: C-terminus) [4,5]. The OD region contains hydrophobic residues necessary for oligomerization and for interacting with host proteins, hnRNPs, and RNA helicases [4,6]. The ARM region is the RRE-binding domain and includes the nuclear localization signal. The NES region is involved in recruiting host cell proteins that are responsible for nuclear export to the Rev–RRE complex [4,5]. The role of linker regions is much less studied. The N-term and C-term influence Rev stability. Turn—the proline-rich region—is involved in additional Rev–Rev interaction. The ONL is a disordered structure and, as supposed, becomes structured during oligomeric assembly on the RRE and during binding with the host nuclear export factor [4].

In naturally occurring viral isolates, Rev–RRE functional activity can vary significantly; this can be related to differences in Rev sequences [7]. In earlier studies, naturally occurring polymorphisms in the OD and NES regions, which can alter the function of the Rev protein, were described [5,8,9]. In a recent study, it was shown that amino acid differences within OD, but not the ARM or NES regions, determined the level of Rev activity [10]. The Rev C-terminus length varies significantly amongst HIV-1 isolates, from the most common 116 a.a. to 123 a.a., because some forms of Rev have the insertion QSQGTET between residues 95 and 96. This insertion has functional value because it reduces Rev’s export activity [4]. Additionally, the presence of inter-subtype differences was noted in the C-terminus. Subtype C contains a prematurely truncated Rev: Rev-C is 16 amino acids shorter than Rev-B [5,11]. Moreover, the results of some studies led to the suggestion of an association between natural polymorphisms in Rev and the clinical status of HIV infection; in particular, that slower HIV disease progression may be attributed to an altered Rev protein [5,12,13].

Genetic diversity is one of the key characteristics of HIV-1 [14]. HIV-1 viruses are classified into the following groups: M (major), O (outlier), N (non-M, non-O), and P. Viruses in group M caused the global HIV epidemic [15,16]. Group M is divided into subtypes: A (sub-subtypes A1–A8), B, C, D, F (sub-subtypes F1–F2), G, H, J, K [16,17,18], and L [19]. Moreover, a large number of recombinant forms between subtypes arise, and are classified into circulating recombinant forms (CRFs) and unique recombinant forms (URFs) [15,20]. CRFs are recombinant viruses isolated from more than three epidemiologically unrelated individuals. URFs are recombinant viruses sampled from only one multiply-infected individual [15,21]. HIV-1 subtypes and recombinant forms are distributed extremely unevenly across the world [15,20]. HIV-1 subtype C is the most widespread variant in the world; it causes more than 50% of all infections and predominantly circulates in Southern and East Africa, as well as in Asia. Subtype B is the most widespread virus variant in North America, Europe, and Australia and is the most studied variant. Subtype A is found in East Africa, Asia, and Eurasia and predominates in Russia and other countries in the former Soviet Union. An increase in the proportion of recombinants is currently being observed. In some countries, recombinant forms are the most common variants: in China, CRF01_AE and CRF07_BC, and in Kyrgyzstan, CRF02_AG [15,20]. Wherein the genetic distance across HIV-1 subtypes is usually 25–35% and that within subtypes is 15–20%, the variation in some areas of the genome, for example in the *env* gene, is higher, while in others, for example the *pol* and *gag* genes, are lower [15,22]. 

HIV-1 variants are characterized by different natural polymorphisms, with single substitutions occurring in ≥1% of cases [23]. It has been suggested that some polymorphisms in HIV proteins that are targets for antiretroviral therapy (protease, reverse transcriptase, integrase) can affect the occurrence of resistance to antiretroviral drugs and can be associated with disease progression [22,24]. Lenacapavir, a first-in-class investigational selective inhibitor of HIV capsid protein, was developed recently [25]. Therefore, HIV capsid protein (p24) conservation in HIV-1 groups M, N, O, and P, including 9 subtypes, 7 sub-subtypes, and 109 CRFs, was evaluated. The results demonstrated that in almost all studied variants, conservation of the sequences was more than 80%, and natural resistance to lenacapavir was unlikely. However, 14 unique markers—natural polymorphisms—in 9 HIV-1 variants in group M were detected, and the researchers concluded that further studies were required to evaluate the impact of genetic polymorphisms in p24 on its functions [26]. Naturally occurring polymorphisms in regulatory proteins are also being actively studied. It has been shown that some polymorphisms in the Tat protein can affect the propagation of the inflammatory signal and neuroinflammation and are subtype-specific [27,28]. Nevertheless, a large-scale study of genetic polymorphisms in the Rev protein in different HIV-1 variants has not been conducted.

The aim of this study was to evaluate the genetic diversity of the Rev protein in different HIV-1 variants in group M through analysis of the sequences available in the Los Alamos National Laboratory HIV Sequence Database.

## 2. Materials and Methods

### 2.1. Strategy for Analysis

In this study, consensus sequences of the Rev protein in different clades of HIV-1 group M were generated and compared. In each of the HIV-1 clades (subtypes or CRFs), the level of amino acid (aa) changes, aa diversity, and the protein variability index were calculated and compared in each Rev residue and each region of its domains. Also, aa conservation and changes specific to individual HIV-1 clades (so-called clade-specific amino acid substitutions, CSSs) were calculated, as described in the study by Troyano-Hernaez et al. [26]. 

### 2.2. Sequence Dataset Compilation

The complete coding regions of HIV-1 Rev protein sequences in amino acids (aa) were downloaded from the Los Alamos National Laboratory HIV Database (www.hiv.lanl.gov/, accessed on 15 September 2023). The analysis focused on HIV-1 group M sequences; URFs were not included in the current study. Sequence selection was ensured by including only nearly full-length HIV-1 genomes and excluding duplicates (only one sequence per patient), sequences with stop codons (within the limits of Rev sequences to be expressed), and more than 5.0% ambiguous amino acids. The retrieved amino acid sequences were aligned against the HXB2 reference (GenBank accession number K03455) using the ClustalX algorithm and analyzed and trimmed in the MEGA v6.0 software [29]; sequence alignments were further curated manually. As a result, a total of 4962 Rev sequences from 5052 nearly full-length HIV-1 genomes with known collection dates between 1984 and 2023 were retrieved. The GenBank Accession numbers, origin information, and sampling dates of the HIV-1 sequences used in this study are presented in Appendix A. 

### 2.3. Consensus Calculation

Consensus sequence calculations were performed using the Consensus Maker tool available at https://www.hiv.lanl.gov/content/sequence/CONSENSUS/consensus.html (accessed on 15 September 2023), using all downloaded sequences per HIV-1 clade. More than three sequences were used per HIV-1 clade to generate consensus sequences, making it possible to assign the most common amino acid state to each site, with no gap removal. The Group M consensus was calculated using all HIV-1 clade consensuses.

### 2.4. Amino Acid Frequencies and Diversity 

The VESPA tool, available at https://www.hiv.lanl.gov/content/sequence/VESPA/vespa.html (accessed on 15 September 2023), was used to calculate the frequency of each amino acid at each position in an alignment. It compares two groups of sequences and looks for a “signature” pattern or the set of amino acids that are conserved in each set but differ between sets, as well as determines distinguishing amino acids and calculates their frequencies in each set. The consensus sequence was used as a background set for intra-set analysis. The amino acid divergence (substitutions per site) in each region was calculated in MEGA5 using the Jones–Taylor–Thornton (JTT) matrix-based model plus the Gamma model. Pairwise alignments were ignored in insertions and deletions of individual HIV-1 sequences that were not included in a consensus.

### 2.5. Calculation of the Variability Index

In addition, the protein variability index was calculated using the Wu–Kabat variability coefficient of the aligned protein sequences, both in the overall dataset including all the sequences and within HIV-1 clades. The WK coefficient was computed following the expression: V_wk_ = N × k/n, where N is the number of sequences in the alignment, k is the number of different amino acids at a given position, and n is the absolute frequency of the most common amino acid at that position [30]. The Wu–Kabat variability coefficient is a well-established descriptor of the susceptibility of an amino acid position to evolutionary replacements, where a WK value of more than 1 indicates the relative variability of the respective site (the higher the value, the greater the diversity).

### 2.6. Statistical Analysis

This study reports the descriptive statistics of amino acid changes and diversity of the Rev protein within individual HIV-1 group M clades. Continuous and categorical variables are presented as medians and interquartile ranges (IQRs) and numbers and percentages (%), respectively. Pairwise comparisons of the amino acid diversity and amino acid conservation levels were conducted using the Mann–Whitney U-test and Fisher’s exact test with Bonferroni multiple-test correction (*p* = α/m, with α = 0.05, m = 650 tests), respectively. For all tests, *p* < 0.05 was considered statistically significant. Statistical analyses were performed using the STATISTICA v.10.0 software (StatSoft, Austin, TX, USA).

## 3. Results

In total, 145 HIV-1 clades (subtypes or CRFs) were sampled based on complete Rev protein sequences in each such set. The numbers of sequences available for each subtype/CRF are described in Appendix A. Of those, 26 subtypes/CRFs with more than 10 sequences allowed for comparison of consensus sequences (Table 1).

### 3.1. Amino Acid Variability in HIV-1 Group M Clades

Although amino acid (aa) changes were analyzed for each of the 26 HIV-1 clades (Table 1), here we will confine ourselves to well-sampled clades with more than 70 sequences (A1, B, C, D, F1, G, 01_AE and 02_AG). All of these clades revealed a total of 52,549 aa changes, 3899 deletions, and 249 insertions (Table 1). The A6 subtype showed the lowest aa changes (9.6 aa changes per sequence), while subtype G and 02_AG were the most change-prone subtypes (15.6 and 15.4 aa changes per sequence, respectively). Variable aa positions ranged from 70.7% in F1 to 98.3% in subtype B; three HIV-1 clades (B, C, and 01_AE) contained more than 90.0% variable positions along their sequences (Table 1). At the level of amino acid diversity in the Rev protein, the A6 subtype (median 12.8 × 10^−2^ substitutions/site) was the most conserved; subtypes B (median 22.9 × 10^−2^ substitutions/site), G (median 21.5 × 10^−2^ substitutions/site), and A1 (median 19.0 × 10^−2^ substitutions/site) varied the most (Figure 1). Regarding the aa changes and the percentage of variable positions in individual Rev domains, these indicators varied greatly across HIV-1 group M clades: from a minimum in the ARM of subtype F1 (0.2 aa changes per sequence and 40.0% variable positions) to a maximum in the C-term domain of subtype A1 (7.0 aa changes per sequence, 100.0% variable positions). The largest aa diversity is observed in the C-term of the C subtype (28.0 × 10^−2^ substitutions/site), while the lowest diversity is observed in the ARM of the D subtype (0 substitutions/site) (Appendix A).

### 3.2. Changes in HIV-1 Consensus Sequences

Heterogeneity in consensus sequences was observed at 72 (58.5%) positions across clades, while the other sites contained the same aa in the consensus sequences, regardless of the level of aa conservation (Figure 2). The ARM domain contained the lowest number of variable positions (26.7%), followed by the N-term (41.6%). Moreover, the ARM domain had the largest number of positions (11) sharing the same aa residue in all the HIV-1 clade consensus: Q36, A37, R38, 40N, R42, R43, R44, W45, R46, R48, and Q49. The greatest number of variable positions was observed in the OD2 domain (80.0%) and the C-term (77.5%), with site 108 (V108A/T/G/D/I/S/P) being the most polymorphic across the HIV-1 group M clades (Figure 2).

The mean Rev amino acid conservation in the HIV-1 group M consensus was 80.8% (Figure 3A); among well-sampled HIV-1 clades with more than 70 sequences available (Table 1), subtypes D (91.3%) and A6 (91.2%) showed the greatest conservation, while subtypes B and G (87.0%) showed the lowest conservation. In total, almost half of the HIV-1 clades contained less than 90 percent mean conservation along their sequences. The most conserved domain in the HIV-1 group M clade was the ARM (93.8%), with 73.3% of its positions showing more than 90 percent conservation (Figure 3B); the maximum conservation value was 99.9%, found in site 76 (P76, NES). The least conserved region was the C-term (69.9%), with over three quarters (82.5%) of positions showing less than 90% conservation; the smallest conservation value (30.3%) was found in site 30 (S68K/E/P, Turn) and 108 (V68A/T, C-term). 

Although the ARM was the most conserved domain in the whole HIV-1 group M clade, the patterns of aa conservation in the Rev domains varied between HIV-1 clades (Figure 3B); the range of conservation of individual Rev domains across different HIV-1 group M clades is described in Appendix A. The ARM domain was the most conserved domain in subtypes A1, B, C, D, F1, and 02_AG, while the N-term domain was more conserved in subtype 01_AE, and the NES domain was more conserved in subtype A6. The smallest conservation value at the Rev domain was 72.2% and was found in the OD2 of the B subtype.

The insertion of seven amino acids between positions 95 and 96 was detected in 15 of the 26 HIV clades studied (Table 1). Meanwhile, subtypes A1 (5*V), C (6*T and 7*E), 07_BC (6*T and 7*E), 08_BC (2*P, 6*T and 7*E), 35_A1D (5*V), 63_02A6 (6*A), and 85_BC (6*T and 7*E) had amino acid changes both relative to the HIV-1 group M consensus and between themselves (Figure 2).

### 3.3. Clade-Specific Amino Acid Substitutions in the Rev Protein of HIV-1 Group M

All aa changes present in >70% of the total sequences of the Rev protein were analyzed, finding 32 single aa CSSs present in 16 of the 26 HIV-1 clades studied; the median CSS was 2. The CSS prevalence and their locations are described in Figure 4. CRF_42BF had the largest number of CSSs–six aa substitutions located in the N-term (11T), OD1 (21R), OD2 (53Q and 54T), and ONL (61N and 74H), while about one-third of the clades (31.3%) studied contained only one aa CSS: A6 had 41Q (74% prevalence), F1 had 95E (82%), F2 had 18Y (93%), 06_cpx had 14L (82%), and 07_BC had 84S (87%). The greatest number of CSSs (13) was recorded in the C-term domain of Rev, and the smallest in the ARM (1); the Turn and NES domains did not contain such substitutions.

### 3.4. Wu–Kabat Protein Variability Index of the Rev Protein in the HIV-1 Group M

Among the 4962 analyzed sequences, 100% of the Rev protein positions had a WK over 1; the median WK was 10.9 (Figure 5). The maximum WK value was 50.4, found in site 14 (K14R/Q/T/L/S/N) located in the OD1 domain, followed by site 108 in C-term (WK = 47.4), site 53 in OD2 (WK = 40.8), site 54 in the OD (WK = 38.1), and site 99 in the C-term (WK = 36.9). A high coefficient value was also recorded for site 82 (WK = 34.5; NES) involved in the nuclear export of Rev. The smallest WK coefficient was 2.0, found in site 76 (NES), followed by sites 2 and 3 in the N-term (WK 3.0 and 4.0, respectively), and sites 19 (OD), 66 (ONL), and 75 (NES), with a WK of 4.0. The ARM (15 aa) within the Rev protein had the lowest median WK of 7.2, with a maximum coefficient of 19.5 in site 47 (A47E/Q/K/R). OD1 (14 aa) showed slightly less variability than OD2 (10 aa), with a median WK of 10.1 vs. 12.4; the N-term domain (12 aa) was more conserved than the C-term domain (40 or 24 aa, depending on the HIV-1 clade), with a median WK of 8.7 vs. 14.2. The median WK for the Turn domain was 8.1 compared with 10.9 for the NES domain and 10.8 for the ONL domain. Regarding the insertion of seven amino acids between positions 95 and 96 within the C-term domain, the maximum WK was 25.8, found in site 7* (T2*E/A), while the smallest WK was 8.0, found in site 4* (G3*R) (Figure 5).

## 4. Discussion

The question of the association between the genetic features of some HIV clades and clinical characteristics has been discussed for many years. Several studies conducted in Africa showed that people infected with subtype D had a higher risk of progressing to AIDS and death than people infected with subtype A [31]. Furthermore, CRF01_AE and CRF75_BF1 HIV infections were associated with lower CD4 counts at baseline and HIV infections induced by CRF01_AE were associated with a higher baseline viral load compared with other co-circulating subtypes [32,33]. For a better understanding of the possible impact of HIV variability on virus transmissibility, replication, and disease progression, recent studies have analyzed the diversity of the *pol* gene and a fragment of the *gag* gene, encoding the p24 protein, among HIV-1 variants. This descriptive study analyzes the Rev diversity among HIV-1 group M clades.

First, the amino acid variability (changes per sequence) and the average amino acid diversity of the full-length Rev protein in the HIV-1 group M clades were evaluated. The A6 subtype had the lowest aa changes and the minimum average amino acid diversity that could be associated with epidemic growth rate. It was shown that the evolutionary rate of A6 variants in the former Soviet Union (FSU) in the fast spread among IDUs was lower than in the slow spread among the heterosexual population [34]. At the same time, mixed epidemics (for example, CRF 02_AG and A1) had greater aa changes and higher amino acid diversity in the full-length Rev protein [35].

Second, the changes in HIV-1 consensus sequences were analyzed, allowing a better understanding of structural, functional, and immunogenic potential differences across HIV-1 clades [36]. As was expected, subtype C had a premature stop codon in position 101 [5,11]. CRF07_BC, CRF08_BC, and CRF85_BC also contained premature stop codons in position 101 (with reference to HXB2) because in this region these variants correspond to subtype C (https://www.hiv.lanl.gov/components/sequence/HIV/crfdb/crfs.comp, accessed on 15 September 2023). F1 and CRF63_02A6 clades had 18T substitution, which was previously associated with the modulation of Rev activity that may delay the onset of AIDS [8]. An interesting finding was the insertion of seven amino acids between positions 95 and 96, which was present in 15 of the 26 HIV clades studied; 8 (A6, G, 01_AE, 02_AG, 13_cpx, 14_BG, 22_01A1, 103_01B clades) of the 15 clades contained a functionally significant insertion, ^95^QSQGTET^96^, which reduces export activity, with different levels of aa conservation. The other seven clades (A1, C, 07_BC, 08_BC, 35_a1D, 63_02A6, 85_BC) contained different variations of this insertion and their functional significance is unknown.

In the third part of the study, the conservation of the Rev protein in group M clades was evaluated. This was found to be 80.8%, which was lower than in protease (PR), reverse transcriptase (RT), integrase (IN), and p24, which have mean values of 93%, 94%, 96%, and 93.6%, respectively [26,36]. This could be explained by overlapping the *rev* gene with the *env* gene, which is known to cause the most striking changes on diversity [4,37]. The ARM domain was expected to be the most well-conserved region of the Rev protein [38]. This agrees with the results of a competitive deep mutational scanning study in which most residues in the ARM domain showed strong selection [4]. Thus, it can be assumed that a well-conserved ARM structure is necessary for Rev function. The N-term also showed a high conservation rate, which could be explained by overlapping with the functionally significant domain in the Tat protein, the arginine-rich motive [4,39]. However, it should be noted that removal of the first 10 residues from the N-term in Rev has no effect on activity [4]. The C-term and OD2 domains were the least conservative regions, as originally suggested by mutational plasticity analysis [4].

Interesting data were obtained in this study when individual aa positions with conservation greater than 70% were assessed. It was found that 16 of 26 HIV-1 group M clades contained a total of 32 aa CSSs. The CSSs were mostly located in the C-term domain and had different frequencies in the N-term, OD, and ONL regions; however, in the ARM region, only one clade (A6) had one CSS—41Q. These data will help in the study of Rev functional activity in different HIV-1 isolates.

In the final part of this study, the WK protein variability coefficient of HIV-1 group M was analyzed to study the susceptibility of each aa position to evolutionary replacement. The higher WK values were located in the OD1 and OD2 domains and the C-term, which is consistent with the results of earlier studies of mutational plasticity in different regions of Rev [4]. However, it is important to note the level of variability in position 82 (34.5) in the NES region. The NES region is involved in Rev interaction with the host nuclear export factor Crm1 [4]; consequently, the variability in this position could have functional significance and requires additional study. Additionally, the variability in residue 51 (20.8) also requires further investigation because Q51 was recently implicated in stabilizing an RNA-bound Rev dimer [4].

Some comments should be made concerning the limitations of this study. First, the small sample size of some HIV-1 clades might not be representative of their real population. Second, the study included only 26 HIV-1 group M clades because of a lack of data for the remaining subtypes and CRFs. These circumstances, combined with the large sequence number variability in each clade, may, in particular, bias the assessment of Rev protein diversity and prevent the detection of all clade-specific aa substitutions within that group. The use of the large number of available partial sequences of the Rev protein could provide a more accurate inference; however, there is a great risk of erroneous estimates due to incorrect subtype classification of partial sequences. Another limitation of this study may be the underestimation of the share of “convenience samples” (received during the local outbreaks of HIV-1, from highly connected transmission networks, etc.) in the considered sequences dataset, which can introduce a downward bias on estimates of the Rev diversity.

## 5. Conclusions

The provided information on Rev protein diversity in the HIV-1 group M clades can be helpful for understanding the association between the genetic features of some HIV clades and clinical characteristics. Generally, the conservation observed in the Rev protein was less than in earlier studies on PR, RT, IN, and p24. Clades G, CRF 02_AG, A1, and B had the highest aa changes, while subtypes B, G, and A1 had the maximum average amino acid diversity. A substitution in position 18, which can supposedly influence HIV infection progression, was detected in clades F1 and CRF63_02A6. In 15 out of 26 studied HIV clades in the consensus sequences, the insertion of seven amino acids between positions 95 and 96 was detected; 8 out of the 15 clades had the insertion 95QSQGTET96, associated with a reduction in the export activity of the Rev protein. A total of 32 single amino acid CSSs were found in 16 clades, while clade A6 had a 41Q substitution in the functionally significant region of the Rev protein. Additionally, in the HIV-1 group M clades, variability was identified in 82 and 51 sites located on the Rev interaction surface. Thus, this study revealed the signatures in the Rev protein in HIV-1 group M clades that can influence Rev activity and clinical characteristics and require further investigation.

## Figures and Tables

**Figure 1 viruses-16-00759-f001:**
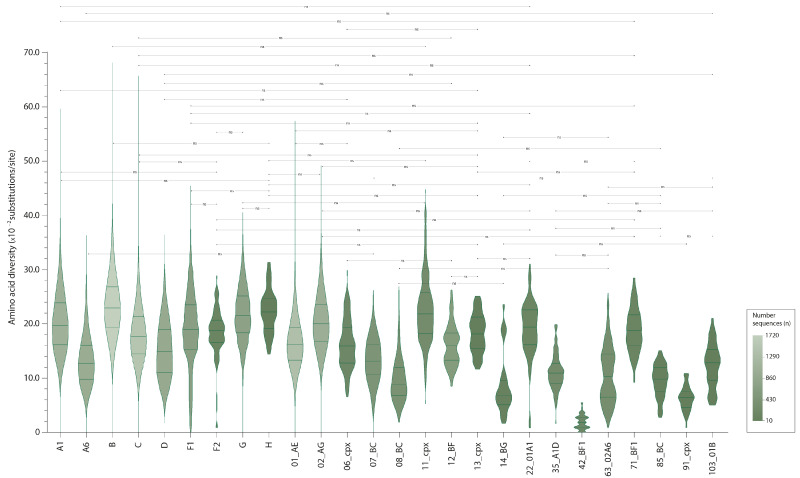
Average amino acid diversity in the full-length Rev protein within individual HIV-1 group M clades. Violin plots show the distribution of amino acid diversity between individual HIV-1 sequences within different HIV-1 clades. Violin plot margins show the distribution of possible values, 25% (Q1) and 75% (Q3) quantiles (IQR), and median values are depicted by horizontal lines; whiskers indicate the 1–99 percentile. The numbers of sequences used in the analysis are displayed using a green color gradient, as described in the legend. The HIV-1 group M consensus was inferred using the 145 clades consensus. *p*-values for pairwise comparisons of the amino acid diversity within different HIV-1 clades assessed using the Mann–Whitney U-test with Bonferroni multiple-test correction (*p* = α/m, with α = 0.05, m = 650 tests) are indicated in Appendix A; the chart shows non-significant (ns, *p* > 0.05) differences only.

**Figure 2 viruses-16-00759-f002:**
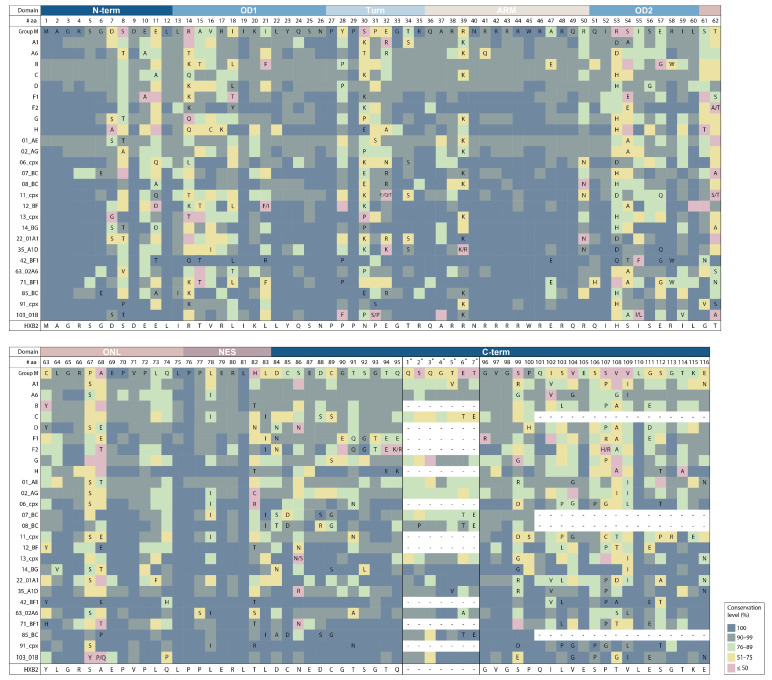
Multiple sequence alignment of consensus sequences of the full-length Rev protein of the HIV-1 group M clades. The amino acids for each position in individual HIV-1 clades (subtypes or CRFs) are with reference to the HIV-1 group M consensus that is inferred using the 145 clades consensus. The consensus HIV-1 group M amino acid for each position is shown directly under the coordinate bar. Amino acids are numbered according to the HXB2 subtype B reference strain; the HXB2 reference sequence is shown below. In the encoding system, discontinuous numbering is used with/without taking into account insertions of seven amino acids between positions 95 and 96; the positions of insertions are marked with an asterisk. The amino acids are colored based on their conservation levels, as indicated in the legend. The matrix of *p*-values for the pairwise comparisons of the Rev amino acid residues with conservation > 91% within different HIV-1 clades assessed using Fisher’s exact test with Bonferroni multiple-test correction (*p* = α/m, with α = 0.05, m = 650 tests) is given in Appendix A. Empty cells represent sequence identity with the consensus; amino acid changes relative to the consensus are indicated using single-letter codes. Annotated protein domains are indicated as colored rectangles: N-term, N-terminal region; OD, Oligomerization domain; ARM, Arginine-rich motif; NES, Nuclear export signal; C-term, C-terminal region; ONL, OD-NES linker.

**Figure 3 viruses-16-00759-f003:**
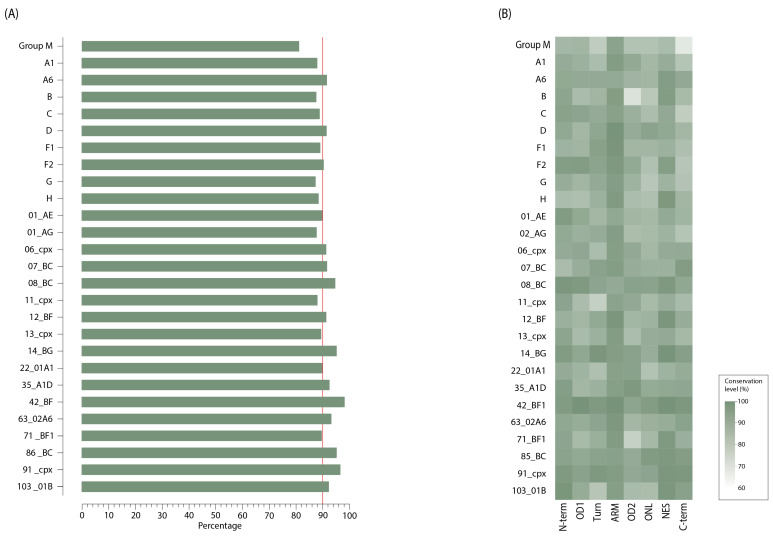
Conservation of amino acid residues of full-length Rev protein sequences across different HIV-1 group M clades. (**A**) Overall percentage of aa conservation in the whole Rev protein; (**B**) percentage of aa conservation for each domain of Rev. Different conservation levels in (**B**) are displayed in different squares using a green color gradient, as described in the legend. The HIV-1 group M consensus was inferred using the 145 clades consensus.

**Figure 4 viruses-16-00759-f004:**
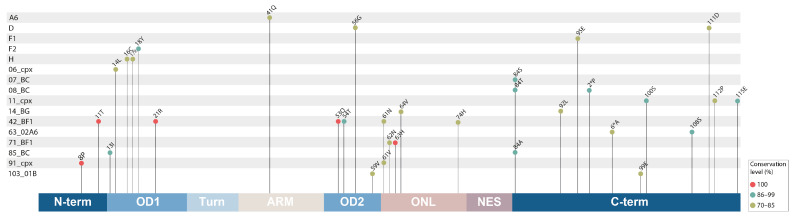
Single clade-specific amino acid substitutions in the full-length Rev protein of HIV-1 group M. The diagram shows all 32 single clade-specific aa substitutions (CSS), the HIV-1 clades they belong to, and their conservations value, which match the color of circles as indicated in the legend; the amino acids are numbered according to HXB2 subtype B reference strain. Amino acid substitutions relative to the consensus are indicated using single-letter codes; amino acid codes are presented in the Figure 3 caption. The digit before the letter shows the position of substitutions in the full-length Rev protein domains (e.g., ‘8P’ indicates the Rev position 8 (N-term domain) and the corresponding amino acid—Proline (P)). Rev domain abbreviations are presented in the Figure 2 caption.

**Figure 5 viruses-16-00759-f005:**
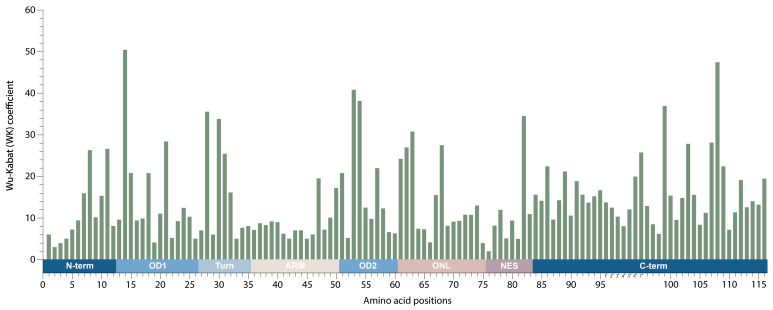
Amino acid variability landscape of HIV-1 group M full-length Rev protein. Rectangular bars represent the Wu–Kabat protein variability index for each of the 116 amino acid residues in the Rev protein; the amino acids are *numbered* according to the *HXB2* subtype B reference strain. In the encoding system, discontinuous numbering is used, taking into account the insertion of seven amino acids between positions 95 and 96; the positions of the insertions are marked with an asterisk. Annotated protein domains are indicated as colored rectangles (above the scale): N-term, N-terminal region; OD, Oligomerization domain; ARM, Arginine-rich motif; NES, Nuclear export signal; C-term, C-terminal region; ONL, OD-NES linker.

**Table 1 viruses-16-00759-t001:** Natural variations in the full-length Rev protein within individual HIV-1 group M clades.

Clade ^a^	Number of HIV-1 near Full-Length Genomes in the HIV Database	Number of Downloaded HIV-1 Sequences Used in Consensus	Number of Changes			Mean Changes per Sequence ^c^	Variable Positions (%)
			Insertions (95→96 Site) ^b^	Deletions	Substitutions		
A1	805	245	7 (*)	322	3374	15.1	89.4
A6	222	187	6 (*)	136	1664	9.6	82.1
B	10,974	1716	61	700	24,897	14.9	98.3
C	2420	865	58 (*)	1435	9074	12.1	97.2
D	219	156	6	23	1534	10.0	79.3
F1	79	74	29	4	942	12.8	70.7
F2	14	14	2	2	160	11.4	48.3
G	101	91	4 (*)	78	1343	15.6	75.6
H	10	10	1	0	135	13.5	46.5
01_AE	2105	613	65 (*)	777	6999	12.7	96.7
02_AG	232	204	13 (*)	424	2722	15.4	87.0
06_cpx	25	17	0	6	167	10.2	47.4
07_BC	46	40	0 (*)	39	327	9.1	57.0
08_BC	37	33	0 (*)	8	191	6.0	49.5
11_cpx	25	23	27	1	323	14.1	64.6
12_BF	14	14	4	4	141	10.3	40.5
13_cpx	10	10	1 (*)	9	125	13.4	39.8
14_BG	14	12	0 (*)	0	68	5.7	29.3
22_01A1	21	15	0 (*)	5	188	12.9	49.6
35_A1D	22	22	0 (*)	14	172	8.4	43.1
42_BF1	17	17	0	6	17	1.3	8.6
63_02A6	26	26	1 (*)	7	187	7.5	47.1
71_BF1	14	14	0	0	163	11.6	44.8
85_BC	11	11	1 (*)	1	59	5.4	32.7
91_cpx	10	10	0	0	34	3.4	21.5
103_01B	10	10	0 (*)	4	88	8.8	30.9

^a^ Only HIV-1 subtypes or CRFs with more than 10 genomic sequences are listed. ^b^ HIV-1 subtypes or CRFs with insertions of seven amino acids between positions 95 and 96 of the Rev protein are marked with an asterisk. ^c^ Includes aa changes and deletions.

## Data Availability

The data that support the findings of this article are openly available at PubMed (https://pubmed.ncbi.nlm.nih.gov/ accessed on 15 September 2024).

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
