# Peer review of "Rev Protein Diversity in HIV-1 Group M Clades"

_viruses, 2024, doi:10.3390/v16050759_

Round 1

Reviewer 1 Report

Comments and Suggestions for Authors

Lebedev and colleagues provide an in-depth analysis of Rev protein sequence diversity based on the LANL sequence data set. Although the work does not provide a lot of new insight, it constitutes a good reference for specialists working on Rev diversity.

I recommend to accept the paper for publication in Viruses, after some minor aspects have been addressed:

- The priority claim (1) in the Abstract (line 27, sentence "This ist the first...") must be removed. There have been analyses on Rev diversity before (see e.g. Table 2 in Li et al., 2015, PMID 25808207). Although the authors provide a lot more detail here, it would always be possible to add another increment of detail to again claim a narrower priority, which is not good practice. Accordingly, (2) the sentence in ll. 374/375 should be removed.
- Figure 4 has the wrong figure legend: the provided text is identical to Figure 3. Please provide the correct one.
- In supplementary tables S3 and S5, values that are considered significant should be highlighted in color. What implication does it have that certain pairs are significantly different?

Minor formal aspects:
l. 40: "non-structural" should be replaced by "accessory" (Vpu is incorporated into particles and can thus be considered as structural protein)
l. 77: font size wrong
l. 104: NaturalLY
l. 106: neuroinflammation
Suppl. Table 1: sheets 2 and 3 have unclear content
Table 1: penultimate column: Headline duplicated
l. 247 to 250: I don't think it is necessary to explain the one letter code for amino acids
l. 256: something missing after "domain of". brackets in the following sentence unclear
Labelling of domains (N-term, OD1 etc.) is hard to decipher on the lightly colored background (Fig. 2, 4 and 5). Use black font on light background.
l. 358: what does "functional cure insertion" mean?
l. 365: "revertase" -> "reverse transcriptase"

Author Response

Response to Reviewer 1 Comments

The authors express their deep gratitude to the expert for your interest in our work. We appreciate the time and effort that you dedicated to providing feedback on our manuscript and are grateful for your comments, which help us to improve our manuscript. We tried to consider all amendments and we hope that the article became clearer and better formulated. Please find the detailed responses below and the corresponding revisions/corrections highlighted/in track changes in the re-submitted files.

Reviewer 2 Report

Comments and Suggestions for Authors

This study describes sequence variability in the Rev protein among HIV-1 group M isolates. The Rev-RRE regulatory axis is essential for HIV replication, but studies of this system have been neglected. A survey of Rev variation is thus needed and timely, and work like this could inform future work on viral pathogenesis and the development of Rev-RRE based therapies. While the authors’ efforts are appreciated, I think there are two major difficulties with this manuscript that should be addressed.

First, in section 3.1 of the results (and elsewhere) the calculations of sequence conservation by subtype would be expected to be highly influenced by the number and type of sequences obtained, especially for subtypes with few sequences available such as 42_BF1. The authors discuss this cogently at lines 333-336 and 390-399, including for 42_BF1 in particular. The fact that all of the 42_BF1 sequences included in the analysis originated from a singular geographically isolated outbreak is not a mere limitation of the authors’ analysis; rather no conclusions should be drawn about Rev diversity in the 42_BF1 subtype at all using the study methodology. It is misleading to discuss amino acid variability in poorly sampled clades at all for the reasons that the authors note themselves. The problem may be less acute for highly sampled clades like B, C, and CRF 01_AE, but the extent to which these diversity calculations are due to artefacts of sampling is unknown outside of 42_BF1. The authors should be very cautious in drawing any conclusions about Rev variability across subtypes for this reason. I would suggest that the authors limit comments of this type to highly sampled subtypes, and perhaps should establish a higher threshold than 10 genomic sequences for inclusion of a subtype in this manuscript.

Second, I have significant concerns about the analysis presented in Figure 2. The subtype B consensus presented here contains 116 amino acids – matching the HXB2 reference which is a subtype B isolate. The subtype C consensus includes 124 amino acids – the 116 included in subtype B plus an additional eight insertions. But this is incorrect: the expressed subtype C Rev is actually shorter than subtype B Rev because the sequence includes a stop codon at HXB2 site 101 and thus contains 107 amino acids (100 from the HXB2 alignment plus a 7 aa insertion). In Figure 2, this stop codon in the subtype C alignment is represented as “-“ at position 101, with additional amino acids that are not actually included in the expressed subtype C Rev shown downstream to position 116. This error would affect the authors’ calculations of amino acid variability at C terminus sites not actually expressed for subtype C and other subtypes with stop codons before site 117 (e.g. 07_BC, 08_BC). It seems as though the authors do not recognize the significance of this stop codon, as at lines 361-363 they discuss the functional significance of an “insertion” between positions 114-115 in clade C. The functional significance of this sequence insertion can be nothing as it is not expressed. I believe that this error will necessitate recalculation of many of the statistics presented in this paper. This error is disappointing, as otherwise Figure 2 is interesting and a meaningful contribution to the field.

I have several less minor comments for the authors’ consideration as well:

·       Regarding the methods, were sequences taken from the Los Alamos Sequence database (line 17, line 112) or NBCI (line 124-125)? Granted, the Los Alamos sequences are found in Genbank, but clarity and a discussion of the search strategy would be helpful for replication.

·       It is not clear how subtype assignment was made for each sequence included in the study. Where annotations of sequence subtype in GenBank accepted at face value? This study would be more robust if sequence typing were performed de novo from the full genomic sequence prior to further analysis.

·       The method for excluding Rev sequences with stop codons (line 128) should be clarified. For example, if a 116 amino acid Rev is taken as the prototype standard, then most subtype C Rev sequences will be seen to include a stop codon at position 108. These sequences were apparently included in the analysis based on Table S1.

·       I am unclear why fewer HIV sequences were “used in consensus” (Table 1) than the number of full-length genomes available. Presumably, this could be because only a single HIV sequence was used when multiple sequences were obtained from an individual (line 128), but this seems inconsistent with the statement that, “Consensus sequence calculations were performed … using all downloaded sequences per each HIV-1 clade” (lines 137-139). This ambiguity should be clarified.

·       In Table 1 and lines 182-183, I do not see the significance of counting the number of amino acid insertions, deletions, substitutions, and changes in total. These values would be more dependent on the total number of sequences included in the analysis rather than on any biologically relevant fact about the HIV subtypes in question. These data also don’t speak to the robustness of the analysis over and above the number of genomes included.

·       In Table 1, the last two column labels (“Mean Changes per Sequence Mean changes per sequence” and “Mean Changes per Sequence Variable positions (%)”) appear to be either typos or a formatting error.

·       Figure 1 – the comparison p-values are too small to be interpreted in the figure as displayed. These should perhaps be eliminated from the figure and included in tabular format in a supplementary table.

·       Figure 3 legend contains several typos.

·       Line 260 – Figure 4B doesn’t seem to exist. Probably the reference should be to Figure 3B.

·       Line 358 – Referring to this insertion as a “functional cure insertion” does not seem well justified. The decrease in nuclear export associated with this insertion in an in vitro assay does not necessarily translate to a clinically significant finding as suggested by the phrase “functional cure.”

·        Line 365 – Change “revertase” to reverse transcriptase.

Comments on the Quality of English Language

I would suggest additional English language editing for this manuscript. While there are few places where the language is unacceptably ambiguous, many constructions (e.g. at lines 25, 37-38, 58, 59, 66, 88-91, 105-107) are somewhat awkward and fluency could be improved.

Author Response

Response to Reviewer 2 Comments

Respected reviewer thank you very much for your valuable comments. Your notes made a huge improvement in our manuscript.  We deeply revised our manuscript and hope that the article became clearer and better formulated. Please find the detailed responses below and the corresponding revisions/corrections highlighted/in track changes in the re-submitted files.

Round 2

Reviewer 2 Report

Comments and Suggestions for Authors

The response of the authors has adequately addressed the issues identified in my review.